# Multi-Omics Analysis of Glioblastoma Cells’ Sensitivity to Oncolytic Viruses

**DOI:** 10.3390/cancers13215268

**Published:** 2021-10-20

**Authors:** Anastasiya V. Lipatova, Alesya V. Soboleva, Vladimir A. Gorshkov, Julia A. Bubis, Elizaveta M. Solovyeva, George S. Krasnov, Dmitry V. Kochetkov, Pavel O. Vorobyev, Irina Y. Ilina, Sergei A. Moshkovskii, Frank Kjeldsen, Mikhail V. Gorshkov, Peter M. Chumakov, Irina A. Tarasova

**Affiliations:** 1V. A. Engelhardt Institute of Molecular Biology, Russian Academy of Sciences, 119991 Moscow, Russia; alipatove@eimb.ru (A.V.L.); asoboleva@eimb.ru (A.V.S.); gkrasnov@eimb.ru (G.S.K.); dkochet@eimb.ru (D.V.K.); pvorobiev@eimb.ru (P.O.V.); 2Center for Precision Genome Editing and Genetic Technologies for Biomedicine, Engelhardt Institute of Molecular Biology, Russian Academy of Sciences, 119991 Moscow, Russia; 3Department of Biochemistry and Molecular Biology, University of Southern Denmark, 5230 Odense, Denmark; vgor@bmb.sdu.dk (V.A.G.); frankk@bmb.sdu.dk (F.K.); 4V. L. Talrose Institute for Energy Problems of Chemical Physics, N.N. Semenov Federal Research Center for Chemical Physics, Russian Academy of Sciences, 119334 Moscow, Russia; jabubis@chph.ras.ru (J.A.B.); emsolovyeva@chph.ras.ru (E.M.S.); gorshkov@chph.ras.ru (M.V.G.); 5Federal Research and Clinical Center of Physical-Chemical Medicine, 119435 Moscow, Russia; i.ilina@rcpcm.org (I.Y.I.); smosh@rcpcm.org (S.A.M.); 6Department of Biochemistry, Medico-Biological Faculty, Pirogov Russian National Research Medical University, 117997 Moscow, Russia

**Keywords:** glioblastoma, interferon, antiviral mechanisms, oncolytic viruses, multi-omic approaches, pathway identification

## Abstract

**Simple Summary:**

This study aims to uncover the contribution of interferon-dependent antiviral mechanisms preserved in tumor cells to the resistance of glioblastoma multiforme cells to oncolytic viruses. To characterize the functionality of interferon signaling, we used omics profiling and titration-based measurements of cell sensitivity to a panel of viruses of diverse oncolytic potential. This study shows why patient-derived glioblastoma cultures can acquire increased resistance to oncolytic viruses in the presence of interferons and suggests an approach to ranking glioblastoma cells by the acquired resistance. Our findings are important for monitoring the oncolytic potential of viruses to overcome IFN-induced resistance of tumor cells and contribute to successful therapy.

**Abstract:**

Oncolytic viruses have gained momentum in the last decades as a promising tool for cancer treatment. Despite the progress, only a fraction of patients show a positive response to viral therapy. One of the key variable factors contributing to therapy outcomes is interferon-dependent antiviral mechanisms in tumor cells. Here, we evaluated this factor using patient-derived glioblastoma multiforme (GBM) cultures. Cell response to the type I interferons’ (IFNs) stimulation was characterized at mRNA and protein levels. Omics analysis revealed that GBM cells overexpress interferon-stimulated genes (ISGs) and upregulate their proteins, similar to the normal cells. A conserved molecular pattern unambiguously differentiates between the preserved and defective responses. Comparing ISGs’ portraits with titration-based measurements of cell sensitivity to a panel of viruses, the “strength” of IFN-induced resistance acquired by GBM cells was ranked. The study demonstrates that suppressing a single ISG and encoding an essential antiviral protein, does not necessarily increase sensitivity to viruses. Conversely, silencing IFIT3 and PLSCR1 genes in tumor cells can negatively affect the internalization of vesicular stomatitis and Newcastle disease viruses. We present evidence of a complex relationship between the interferon response genes and other factors affecting the sensitivity of tumor cells to viruses.

## 1. Introduction

Glioblastoma multiforme (GBM) is an aggressive form of a primary brain tumor [1]. It remains an incurable disease with a median survival period from 12 to 15 months [1,2,3,4]. Standard therapy, which involves surgical removal followed by radiotherapy in combination with temozolomide, provides only temporary relief and ends up with a secondary tumor growth [5] that usually manifests as a terminal stage in the development of the disease. Tumor-initiating glioblastoma stem cells (GSC) contribute to recurrences as they tend to migrate into the normal brain tissue, thus escaping both radiation treatment and chemotherapy [6,7,8]. Therefore, the development of alternative therapeutic approaches for GBM treatment is of primary importance. Oncolytic virotherapy is one of these promising approaches, demonstrating clinical examples of long-term remission in some patients. Oncolytic viruses (OVs) may directly and selectively kill GBM cells [9], reverse the immunosuppressive microenvironment in the tumor, and boost both innate and adaptive antitumor immunity branches [10,11,12]. OVs can stimulate antitumor immunity by recruiting antigen-presenting cells, natural killers, and cytotoxic T-lymphocytes into the tumor [13,14,15], providing prolonged antitumor effects even after the immune system neutralizes the virus. OVs have also been shown to efficiently target the GBM-initiating stem cells [16,17,18,19,20,21]. Unlike chemotherapy, OVs are well tolerated and cause only minor side effects, including short-term moderate fever and fatigue that is easily manageable by antipyretics. The OVs of different families were tested for GBM treatment, delivered either locally or systemically. Recently, in a study with recombinant poliovirus PVSRIPO applied locally as a single dose infused after surgical removal of the tumor, 20% of GBM patients demonstrated remissions lasting over 72 months, while the other 80% showed no notable lifetime extension [22]. The different therapy responses suggest that GBM cells of some patients are resistant to a particular oncolytic virus. However, the cells’ resistance to the single virus may not preclude their ability to respond to a different therapeutic virus strain. During the malignant progression, GBM cells of patients follow individual passes to acquire autonomous growth. Multiple events of consecutive genetic and epigenetic changes are required for the cells to escape normal growth control and immune system surveillance. Typically, cancer cells tend to lose the innate antiviral mechanisms that include their ability to induce type I interferons and develop the antiviral response [10,23,24,25,26,27,28] as the antiviral mechanisms limit cells’ ability for uncontrolled proliferation. However, other molecular events could affect the ability of cells to support the replication of viruses, such as the loss of mechanisms of virus entry because of the aberrant expression of specific receptors or defects in endocytosis, virus uncoating, etc. Understanding specific determinants affecting tumor cells’ sensitivity to oncolytic viruses is important for predicting the responses of patients to a virus therapy and choosing essential virus strains for personalized treatment. Still, there is a significant lack of available data on the state of antiviral innate immunity mechanisms and other components affecting sensitivity to viruses in individual GBM cells. Approaches that integrate classic molecular biology methods with large-scale transcriptomics and proteomics analysis of GBM cells could provide mechanistic insights explaining the variations of responses of GBM to oncolytic virotherapy.

This study analyzed the individual responses of eight patient-derived early passage GBM cells to treatment with type I interferons (IFNs) by proteomics and transcriptomics analysis. By comparing the data with the cells’ ability to develop the IFN-induced resistance to virus infection, we identify genes that explain the observed differences in virus sensitivity and estimate the contribution of interferon mechanisms to the self-protection of glioblastoma cells against oncolytic viruses.

## 2. Materials and Methods

### 2.1. Cells

Low-passage cultures of GBM cells were prepared from surgically removed tumor tissue obtained from N.N. Burdenko Institute of Neurosurgery (Moscow, Russia). Fresh tumor fragments were disintegrated by passing through 50 micron nylon mesh, washed two times with 30 volumes of DMEM, and seeded to plastic dishes in DMEM with 10% FBS and incubated in 5% CO_2_ 37 °C. Cultures were split when reaching sub-confluency. The cultures were grown for no more than 8–10 passages before being used in the study. Before protein extraction, harvested cells were washed three times with PBS and stored at −80 °C. Normal embryonic astrocytes were obtained from abortive material as described elsewhere [29]. The DBTRG-05MG cell line was obtained from ATCC (CRL-2020™). All cells were authenticated, and the cultures were found free of mycoplasma contamination using the MycoReport test (Evrogen, Moscow).

### 2.2. Viruses

The following virus strains were used: Vesicular stomatitis virus Indiana strain (VSV-I), Sabin vaccine strain of type 3 poliovirus (PV3), Newcastle disease virus strain H2 (NDV-H2), Coxsackie B5 virus strain LEV14 (GenBank: MG642820.1), Coxsackie A7 strain LEV8 (GenBank: JQ041367.1), and Echovirus 12 strain LEV7. The viruses were propagated in RD cells, virus titers (as TCID50/mL) were evaluated in RD cells [30].

### 2.3. Testing the IFN Response Using VSV Assay

For testing the IFN response, GBM cultures seeded at 5000 cells/well in 96-well plates were treated with IFNα-2b (Farmaclon, Russia) at 100, 200, 500, 1000, 2000, and 5000 units/mL, or with IFNβ (Farmaclon, Russia) at 10, 100, and 1000 units/mL and incubated for 24 h. Then, the cells were infected with VSV-I at MOI = 1.0. Cytopathic effect (CPE) was estimated by morphology in 24 and 48 h (Figure 1a). Cell cultures unprotected against VSV were fully killed (visually, only single cells of 5000 population were survived) in 24 h, even if cells were pretreated with high IFN concentration (1000 units/mL). Therefore, cellular responses for VSV assay were summarized in “binary” form: either the cells survived or were killed (Figure 1a). Eight GBM cultures were divided into two groups based on the observed changes in cell morphology. The IFNα- and IFNβ-treated and control samples were subjected to LC-MS/MS-based proteome analysis. Cells were treated for 24 h at 37 °C with IFNα-2b and IFNβ at 100 and 1000 units/mL concentrations, respectively. For the proteome analysis, the sub-confluent cell cultures were grown and treated in 6 cm culture plates. The cells were scraped from the surface, washed three times with cold PBS, pelleted by low-speed centrifugation, and stored in liquid nitrogen until use. In addition, four biological replicates of glioblastoma cultures and normal astrocytes, treated with IFNα-2b at a concentration of 100 units/mL for 24 h, and the untreated controls were propagated and further used for proteome analysis to evaluate the IFNα-induced response specific for each cell culture.

### 2.4. Testing the IFN Response by Titration-Based Measurements of Cell Sensitivity and Virus Replication

For the sensitivity tests of patient-derived GBM cells (Figure 1b), wild-type DBTRG-05MG, and its subcultures, the cells were seeded at a density of 5000 cells/well in 96-well plates and incubated for 24 h at 37 °C in a 5% CO_2_ atmosphere. Next, IFNα-2b was added to the samples at concentrations of 10, 50, 150, 1000 units/mL and incubated for the next 24 h under the same conditions. Then, IFN-treated and untreated (control) cells were infected with VSV-I (Indiana strain) and enterovirus strains at the wide range multiplicity of infection (MOI) from 100 to 0.0001 (10-fold dilutions of stock virus solutions for titration). Cytopathic effects (tissue culture infection dose, lgTCID 50/mL) were estimated after 24 and 72 h following the infection.

For patient-derived cultures, wild-type DBTRG-05MG and its derivatives, virus production was measured. Cells were plated on 60 mm Petri dishes (1 × 10^5^ cells per dish), and cultured overnight at 37 °C in DMEM supplemented with 10% FBS. The next day, cells were treated with recombinant interferon α-2b at 1000 units/mL concentrations in DMEM with 10% FBS. Control samples were incubated in DMEM with 10% FBS. After 24 h, the cells were separately infected with each enterovirus strain and VSV at MOI 1. In 48 h, the supernatant was collected after centrifugation, and the virus titer was determined.

Titration-based measurements of virus production and sensitivity to viruses for DBTRG-05MG, its derivatives, and patient-derived cells were performed in four biological replicates. Statistical evaluation of data was conducted using GraphPad Prism software (v.8.0.2) for Windows (GraphPad Software, La Jolla, CA, USA) using following settings: Unpaired *t*-test, correction for multiple comparisons using Two-stage step-up (Benjamini, Krieger, and Yekutieli) method, desired FDR (Q-value) < 1.00% or 5.00% (for evaluation of IFN-induced protection against viruses in patient-derived cells and DBTRG-05MG derivatives, respectively). The data and results of statistical testing are provided in Appendix A.

### 2.5. Real-Time qPCR

Real-time PCR was performed with SYBR Green qPCR master mix (USB) in a CFX96 Touch™ Real-Time PCR Detection System (Bio-Rad, Hercules, CA, USA). The PCR protocol was as follows: initial activation at 95 °C for 5 min, 40 cycles at 95 °C for 10 s, and 60 °C for 40 s. Ct values were converted into relative gene expression levels compared to the internal control gene, GAPDH. Each PCR run was performed in quadruplicate. The primer sequences are provided in Appendix A. Primers were purchased from DNA-Synthesis Ltd. (Moscow, Russia).

### 2.6. Western Blotting

Cells were lysed with the M-PERTM mammalian protein extraction reagent (Thermo Scientific, Munich, Germany) with a 1% cOmplete^TM^ EDTA-free protease inhibitor cocktail (Roche Diagnostics, Mannheim, Germany). The supernatant was separated by centrifugation at 13,000 rpm for 10 min at 4 °C. Total protein concentration was measured using the Quick Start^TM^ Bradford protein assay kit (Bio-Rad, Hercules, CA, USA). Protein samples were mixed with 5 × Laemmli loading buffer, denatured at 100 °C for 5 min, then cooled and stored at −20 °C until further use. Protein aliquots from each sample (10 μg), along with a protein marker (PageRuler^TM^ Plus pre-stained protein ladder, Thermo Scientific, Munich, Germany) were separated by 12% SDS-PAGE and transferred to a membrane (PVDF, Amersham^TM^ Hybond^®^ P, GE Healthcare) at 80 V for 3 h. The electrophoresis and transfer to PVDF membrane were performed using Mini-PROTEAN Tetra cell (Bio-Rad, Hercules, CA, USA). Membranes were blocked in phosphate-buffered saline with Tween solution (PBST, 137 mM NaCl, 2.7 mM KCl, 10 mM Na_2_HPO_4_, 1.76 mM KH_2_PO_4_, pH 7.4, and 0.1% Tween-20) containing 5.0% of nonfat dry milk for 1 h at room temperature. Blotted membranes were then incubated with primary Rabbit monoclonal antibodies to Oas2, Mx1, Ifit3, Plscr1, Bst2 (Abcam, Waltham, MA, USA) at 1:1000 dilution, overnight at 4 °C, washed three times in PBST for 10 min, and incubated for 1 h at room temperature with secondary HRP-labeled mouse anti-Rabbit antibodies (Santa Cruz Biotechnology, Santa Cruz, CA, USA) at 1:5000 dilution. Finally, blots were rinsed in PBST 3 times for 10 min and visualized using ECLTM Plus Western Blotting Detection System (Amersham^TM^, GE Healthcare, Helsinki, Finland) and ChemiDoc™ Imaging System (Bio-Rad, Hercules, CA, USA).

### 2.7. Gene Knockdown/Overexpression

Stable cell lines with knockdown of selected genes and control cell line, expressing shRNA for GFP2, were generated by lentiviral transduction with the pLSL-puro vector [31]. Lentiviral constructs for hyperexpression were based on the pLCMV-PL3 series vectors [32], expressing target genes under CMV promoter control. Further, the cell lines were passaged several times on puromycin (1.5 µg/mL). The cells were scraped, washed three times with cold PBS, pelletized by low-speed centrifugation, and stored in liquid nitrogen until further use. The quality of shRNA-mediated knockdown/overexpression was verified using standard procedures. Western blots and mRNA measurements using real-time qPCR are provided in Appendix A.

### 2.8. Transcriptomics Data

Publicly available data of differentially expressed IFN-stimulated genes derived from large-scale quantitative transcriptomics study of human skin primary fibroblasts treated with IFNα was used to compare the classic IFN response determined at transcript level with the proteomic responses in GBM cells [33].

Experimental data were obtained at the Genome Center of EIMB RAS, Moscow, Russia. Experiments were performed in triplicate. GBM cells were washed with PBS, followed by lysis using 500 μL of lysis buffer (MagNA Pure Compact RNA Isolation Kit, Roche Diagnostics, Mannheim, Germany). Total RNA was extracted using the RNeasy Mini kit (QIAGEN, Hilden, Germany) according to the manufacturer’s protocol. The quality and quantity of RNA were determined using Agilent 2100 bioanalyzer (Agilent Technologies, Santa Clara, CA, USA). All RNA samples had RINs above 9.5. Total RNA (2 μg) from each sample was used to prepare the mRNA library using the TruSeq RNA Sample Preparation Kit v2 Low Sample (LS) protocol (Illumina, San Diego, CA, USA) according to the manufacturer’s protocol. The cDNA libraries were sequenced (single-end reads, 50 bp) using Illumina NextSeq 500 system. Raw reads were checked for quality and trimmed using the FastQC and Trimmomatic tools [34]. The trimmed reads were mapped to the human reference genome (GRCh38.80, Ensembl annotation) with a splice-aware STAR mapper [35]. Next, 5′-to-3′ transcript read coverage distribution was evaluated using geneBody_coverage.py script from RSeQC package [36] to ensure the absence of 3′-tail bias. Read counting per gene was performed using the featureCounts tool from the Subread package [37]. Further analysis was performed in the R environment using edgeR (identification of differentially expressed genes), topGO, and clusterProfiler (GO, KEGG, Reactome enrichment analyses). The results of the statistical analysis are summarized in Appendix A. The RNA sequencing data is available at NCBI Gene Expression Omnibus (GEO) under the accession number GSE163949.

### 2.9. Cell Lysis and Digestion for LC-MS/MS Proteome Analysis

Cells were resuspended in 100 μL of lysis buffer and stirred for 60 min at 1000 rpm at room temperature. The buffer contained 0.1% *w/v* ProteaseMAX Surfactant (Promega, Madison, WI, USA) in 50 mM ammonium bicarbonate, and 10% *v/v* ACN. Lysis was performed by sonication for 5 min at 30% amplitude on ice (Bandelin Sonopuls HD2070, Berlin, Germany). The supernatant was collected after centrifugation. Total protein concentration was measured using Pierce’s quantitative colorimetric peptide assay (Thermo Scientific, Munich, Germany). Protein extracts were reduced in 10 mM DTT at 56 °C for 20 min and alkylated in 10 mM IAA at room temperature for 30 min in the dark. Then, samples were overnight digested at 37 °C using sequencing grade modified trypsin (Promega, Madison, WI, USA) added at the ratio of 1:50 *w*/*w*. Digestion was terminated by acetic acid (5% *w*/*v*). Samples were stirred (500 rpm) for 30 min at 45 °C followed by centrifugation. The supernatant was collected, passed through the 10 kDa filter unit (Millipore, Burlington, MA, USA), and centrifuged. After that, 100 μL of 50% formic acid was added to the filter unit and centrifuged again. Samples were dried using a vacuum concentrator and stored at −80 °C until further analysis.

Before the analysis, peptides were desalted using cartridges for solid-phase extraction (Oasis HLB, 1 cc, 30 mg, 30 μm particle size, Waters, Milfold, CT, USA). For label-based quantitation, desalted dry samples were resuspended in 100 mM HEPES (pH 8.5) and labeled with the TMT11-plex kit (Thermo Fisher Scientific, San Jose, CA, USA) according to the manufacturer’s protocol.

### 2.10. LC-MS/MS Analysis

LC-MS/MS analysis was performed using Orbitrap Fusion Lumos, QExactive HF and Q-Exactive Plus mass spectrometers (Thermo Scientific, Waltham, MA, USA) coupled with UltiMate 3000 nanoflow LC system (Thermo Scientific, Bremen, Germany). The loaded sample quantity was 1 μg per injection. Trap column μ-Precolumn C18 PepMap100 (5 μm, 300 μm i.d. × 5 mm, 100 Å) (Thermo Scientific), analytical C18 column (1.8 μm, 75 μm i.d. × 300 mm, 100 Å) home-packed with ReproSil ODS-3 2-μm sorbent (GL Sciences) or EASY-Spray PepMap RSLC C18 analytical column (2 μm, 75 μm i.d. × 500 mm, 100 Å) (Thermo Scientific) were employed for separations. The column temperature was at room temperature or set to 50 °C. Mobile phases were: (A) 0.1% FA in water; (B) 95% ACN, 0.1% FA in water or (A) 0.1 *v/v* % FA in water; (B) 80 *v/v* % ACN, 0.1 *v/v* % FA in water. Pre-concentrated peptides were eluted using a linear gradient from 5%B to 20%B for 105 min, followed by a linear gradient to 32%B for 15 min at 270 nL/min flow rate. Another set of employed gradient conditions was a linear gradient from 8%B to 20%B in 65 min followed by a linear gradient to 35%B for 95 min, and then to 55%B in 18 min at 440 nL/min flow rate. Mass spectrometry measurements were performed in data-dependent mode. Precursor ions were isolated in the *m/z* window of 0.7 Th or 1.2 Th followed by HCD fragmentation at a normalized collision energy of 30 or 32. Fragment ions were measured in the Orbitrap mass analyzer with a resolving power of 30,000 or 35,000 at *m/z* 200. Proteomics data are deposited to ProteomeXchange (projects PXD022836 and PXD022906).

### 2.11. Protein Identification and Quantitation

Thermo raw files were converted to mgf or mzML format using msConvert (http://proteowizard.sourceforge.net/tools.shtml, accessed on 17 May 2019). Database search was performed using MSFragger (version MSFragger-20171106) [38] and Identipy [39] against a combined target-decoy database (db). The target db was SwissProt human (access date 18 June 2018); the decoy db was compiled by reversing the protein sequences from the target db. Search parameters were as follows: 10 ppm for precursor mass tolerance; 0.01 Da for fragment mass tolerance; maximum two missed cleavage sites; fixed carboxyamidomethylation of Cys. Settings for potential modifications were optimized based on the results of open search profiling within 200 Da precursor mass tolerance followed by MS/MS-based modification site assignment [40]. Met oxidation (+15.9949 Da), peptide N-terminus formylation (+27.994915 Da) were set as potential modifications. Filtering to 1.0% protein false discovery rate, post-search validation, as well as label-free relative quantitation of identified proteins by NSAF method [41] were performed using Scavager [42]. Before statistical analysis of label-free data, protein abundances (NSAFs) were log-transformed and then normalized to eliminate global technical biases, using the following equation: *log_2_(NSAF_i_)_norm_ = (log_2_(NSAF_i_) − log_2_(NSAF)_mean_)/log_2_(NSAF)_std_*, where *log_2_(NSAF)_mean_* and *log_2_(NSAF)_std_* are the mean and standard deviation of log-transformed NSAFs in the replicate, respectively. Abundances of proteins missed in some replicates were imputed by minimal NSAF observed in the respective LC-MS/MS replicate using a scaling factor of 10^−3^ or 1.0. For label-based quantitation in TMT11plex experiments, Diffacto [43] was adopted for use with intensities of isobaric labels.

### 2.12. Statistical Analysis of Proteomic Data

In this work, we utilized two study designs compiled to characterize different features of GBM response to type I IFNs. The first design (Appendix A) aims to determine the type I IFN response across the cohort of glioblastoma cultures, compiled according to the observed CPE (Figure 1a). The goal of the second design (Appendix A) was to determine culture-specific (personalized) responses to IFNα and its comparison with the IFNα response of normal astrocytes. Choice of statistical tests for label-free quantitation was made based on the standard recommendations [44]. Statistical analysis was performed using ANOVA implemented in Python module statsmodels. The normality of the data was assessed by the Shapiro–Wilk test. Since some population distributions were not normal, a non-parametric alternative to ANONA, the Kruskal–Wallis test, was also applied to the data. ANOVA *p*-values were corrected for multiple comparisons using the Benjamini–Hochberg method [45]. For statistical analysis of label-based proteomic data, the PECA method [46] built-in Diffacto software [43] was used. The PECA values were corrected for multiple comparisons [45]. Results of statistical analyses are provided in Appendix A.

For the GO analysis, proteins were split into two groups: (1) upregulated and (2) downregulated. Each group was evaluated separately using STRING db [47]. STRING db was programmatically accessed using the available API (http://version10.string-db.org/help/api/ accessed on 12 December 2020). To recognize the ISGs, the database of type I interferon-stimulated genes (ISGs) for *Homo sapiens* was downloaded from http://isg.data.cvr.ac.uk/ (accessed on 6 September 2020) [33]. Additionally, protein and gene descriptions deposited at https://www.uniprot.org/ (accessed on 8 October 2020), https://www.genecards.org/ (accessed on 13 October 2020), and http://www.interferome.org/ (accessed on 2 October 2020) were used. KEGG pathway analysis was performed using https://pathview.uncc.edu/ (accessed on 21 January 2021).

### 2.13. Analysis of Cross-Associations between Omics Data and Titration-Based Measurements of Cell Sensitivity and Virus Replication

Consistency of protein abundances (log2-transformed normalized NSAF) in biological and technical replicates was assessed by calculating the pairwise Pearson and Spearman correlation coefficients. Outliers with averaged correlation coefficients below 0.75 were excluded (Appendix A). Welch’s *t*-test was used to assess statistical significance of changes at protein level between IFN-treated and control samples.

To explore cross-associations between data, Pearson and Spearman correlations between transcriptome/proteome-derived Log_2_FC (IFN to control) for each gene/protein and the cell culture sensitivity to VSV-I and VSV-I replication were calculated. Then, data were sorted by both Pearson and Spearman correlation coefficients and cut off by the average absolute value, abs(R) = 0.75 (Appendix A, the sheets named “RNAseq repl.-assoc. gene list”, “RNAseq sens.-assoc. gene list”, “Proteome repl.-assoc. gene list”, and “Proteome sens.-assoc. gene list”). GO enrichment analyses of genes and proteins passed the above cutoff (for which differential expression was highly correlated with cell sensitivity to VSV and VSV replication rates), were performed using the g:Profiler web tool (https://biit.cs.ut.ee/gprofiler/gost, accessed on 20 September 2021). The results of GO analyses are summarized in Appendix A (the sheets named “RNAseq repl.-assoc. GSEA”, “RNAseq sens.-assoc. GSEA”, “Proteome repl.-assoc. GSEA”, “Proteome sens.-assoc. GSEA”).

## 3. Results

### 3.1. GBM Cultures Demonstrate Different Levels of IFN-Induced Protection against VSV-I

To reveal subtle differences in the ability of Type I IFNs to induce the antiviral state in eight GBM cultures obtained from individual patients, we pretreated cells with IFNα or IFNβ before infecting them with VSV-I and some other non-pathogenic viruses: NDV-H, PV3, Echo12, Coxsackie A7, and Coxsackie B5. Figure 1a summarizes the state of the cells after 48 h of VSV-I infection. In five cultures, treatment with IFNα or IFNβ successfully protected cells from VSV-I-induced death, suggesting relatively preserved interferon mechanisms of the antiviral response. In three cultures, notable defects in IFN response were found. Pretreatment with IFNβ was not able to rescue either of the cultures from VSV-I. However, IFNα induced complete or partial protection of GBM5222 and GBM5522 cells, respectively. The GBM4114 cells remained virus-sensitive even upon treatment with IFNα, suggesting a severe deficiency in IFN response. Based on the observed cytopathic effects, we divided GBM cultures into two groups: (1) cultures protected against VSV-I with IFNα and IFNβ; (2) cultures with more profound defects protected by IFNα but not IFNβ, or not protected by either type of IFNs.

GBM cultures were also tested for sensitivity to a variety of non-pathogenic viruses (Figure 1b). The chosen viruses have different abilities to overcome IFN-induced antiviral defense, while malignant cells have different abilities to acquire protection against viruses due to IFN treatment. Figure 1b summarizes the results of virus titration used for quantitative measurements of the cell sensitivity to viruses and virus replication efficiency in the cells. Statistically significant IFN-induced changes in cell sensitivity and virus replication for each culture are highlighted by “stars”. We suggest that the more significant number of “stars” per cell culture is observed, the stronger the IFN-induced defense is acquired by the GBM culture. If cultures have the same number of “stars”, we suggest ranking them based on averaging the statistical significance of “stars”. For example, cultures GBM3821, GBM5410, and GBM6067 have a code of four “stars” each, and their mean Q-values are 0.0053, 0.001912, and 0.000767, respectively (Appendix A, “primary_cultures_statistics”). Other scoring schemes were also tested and resulted in the same rank (Appendix A). The suggested estimate allows the ranking of GBM cultures by the strength of IFNα-induced resistance from the weakest to the strongest: GBM4114, GBM5222, (GBM3821, GBM5410, GBM6067), GBM5522, GBM6138, and GBM4466. We further used comparative proteomics and transcriptomics to identify specific molecular compounds that account for the differences in the IFN-induced antiviral response at the protein and mRNA levels.

### 3.2. A Unique Molecular Portrait of Proteome Changes Differentiate the GBM Cultures from Being Silent or Responding to IFN Stimulation

To identify characteristic molecular patterns in GBM cells’ responses to type I IFNs, we performed a comparative proteomics study using two different statistical designs (Appendix A). The first design (Appendix A) was aimed at determining the cellular response, “averaged” over the “preserved” and “defective” groups of different GBM cultures, according to Figure 1a. The results are provided in Figure 2. Cell pretreatment with IFNα (Figure 2a,c) and IFNβ (Figure 2b,d) led to statistically significant changes in several hundreds of proteins in both compared groups. However, both fold changes and *p*-values for upregulated proteins correlated with the IFN subtype used in the treatment and the sensitivity of the cells to IFN, according to Figure 1a. Our data showed that IFNα treatment more strongly affected a protein regulation compared with IFNβ. To achieve similar responses, the cells were treated with 100 and 1000 units/mL of IFNα and IFNβ, respectively. In cells with preserved antiviral IFN response, most of the altered proteins corresponded to known IFN-responsive antiviral proteins, as well as the proteins involved in the innate immune response (according to the Interferome database [48] and GO analysis using STRING [47] (Appendix A)). Thus, in cultures with a preserved antiviral response and treated with either IFNα or IFNβ, the protein products of the MX1/MX2, IFIT1/IFIT2/IFIT3, OAS2/OAS3/OASL, and BST2 genes were the primary and equally overrepresented antiviral proteins. Other proteins were the products of genes associated with the innate immune system and MHC class I/II antigen processing and presentation, such as RNF213, PARP9/PARP14/DTX3L, CTSS, SAMD9L, HERC6, LGALS9, and DDX58/IFIH1. These genes reportedly promote the induction of IFNα/β pathways (https://www.genecards.org/, accessed on 13 October 2020).

We performed a quantitative proteomics analysis according to the statistical design shown in Appendix A for the individual IFNα responses for (1) normal astrocytes (Appendix A); (2) two GBM cultures with a preserved antiviral response (Appendix A); (3) two GBM cultures with a defective antiviral response (Appendix A). Comparing data from the virus susceptibility tests (Figure 1a,b) and the proteome analyses (Figure 2, Appendix A) suggests that with more robust antiviral protection, higher fold change values and lower *p*-values are observed for protein products of IFN-responsive genes. Notably, all upregulated proteins were recognized as IFN-induced across the public ISG databases [33,48]. GO analysis was performed to ensure enrichment of type I IFN signaling pathways (Appendix A). The GBM cultures protected from VSV-I by IFN treatment demonstrated enrichment on the upregulated proteins of the antiviral response and interferon signaling, including the JAK-STAT cascade.

To find a reproducible molecular portrait revealed by various experimental designs in GBM cultures and normal astrocytes, we hierarchically clustered the IFN-induced fold changes of upregulated proteins (Appendix A). Across different pairwise comparisons, besides the expected differences between the individual and group-averaged responses, we observed a regular co-expression of about 30 proteins.

### 3.3. IFN-Induced Molecular Changes in IFN-Protected GBM Cells Are Similar to Those of the IFNα Response in Normal Human Fibroblasts

To clarify how many of the altered proteins correspond to typically observed ISGs, the protein fold changes after the treatment with IFNα in the IFN-protected (“preserved”) group of GBM cultures (Figure 1a) were compared with the earlier determined transcript fold changes in the similarly treated normal human skin primary fibroblasts [33] (Figure 3a). The comparison revealed that 95% of proteins upregulated in IFNα-sensitive GBM cells have a co-directed expression of the respective genes in the human interferome taken as a reference. Genes and the respective fold changes are given in Appendix A. We believe these datasets are well correlated despite the difference in biological models. This comparison validates our results and confirms that the observed cellular responses to IFNα in the IFN-“preserved” GBM cells are similar to the IFNα responses typically observed in normal human cells.

### 3.4. Transcriptomic and Proteomic Portraits of GBM Cells with Preserved and Defective IFN Responses Coincide

Transcriptomic analysis of GBM cultures (GBM3821, GBM4446, GBM5522, and GBM4114) also revealed quantitative changes in key regulators and members of the JAK-STAT and RIG-I-like receptors’ signaling cascades, which correlated with CPE in the VSV-I sensitivity tests after treatments with IFNs. The strongest co-expression of STAT1, STAT2, IRF9, ISG15, TRIM25, DDX58 (RIG-I), and the others were observed in GBM3821, GBM4446, and GBM5522 cells, developing resistance to VSV-I after IFNα pretreatment. In contrast, in GBM4114 cells, their expression was significantly reduced (Appendix A). Comparing data shown in Appendix A demonstrates that large-scale transcriptome changes for GBM3821 (Appendix A), GBM5522 (Appendix A), and GBM4114 (Appendix A) cells are consistent with the changes in proteomes. Furthermore, the amplitudes of the responses coincide dependently with the magnitude of IFN-induced protection (Figure 3c). Further comparison of the omics responses with the results of the tests on sensitivity to VSV-I and other viruses shown in Figure 1a,b demonstrates that the number of ISGs, their fold changes, and significance unambiguously justify the changes in cell sensitivity to viruses and the efficiency of virus replication in GBM cells. To further confirm the ISG co-expression at protein and mRNA levels, we also compared the changes induced by IFNα in GBM3821 cells (Figure 3b).

We hypothesized that proteins exhibiting the most profound IFN-induced quantitative changes in GBM cells could be among the products of genes that usually determine the resistance of normal cells to viruses in response to treatment with Type I IFNs. To choose target genes, we used results of hierarchic clustering of quantitative changes (with FC > 2.0) across the samples with preserved IFN-induced protection (Appendix A), which integrates “averaged” (Figure 2a,b) and “personalized” (Appendix A) cell responses. Thus, genes’ encoding proteins identified across at least four out of five sample comparisons were considered promising targets. Such a “core” response across the sample comparisons counted about 30 proteins, which we empirically split into two subgroups: (1) well-studied proteins with antiviral activity (MX1/2, IFIT1/2/3, OAS1/2/3/L, etc.), and (2) proteins with understudied roles in response to IFNs (PLSCR1, BST2, PARP9, etc.). Of each subgroup, we randomly selected a few genes: (1) well-studied (IFIT3, OAS2, and MX1), and (2) understudied (PLSCR1 and BST2). The goal was to identify genes whose suppression or overexpression affects GBM cells’ sensitivity to a panel of potentially oncolytic viruses (NDV-H2, PV3, Echo12, Coxsackie A7, and Coxsackie B5) used in this study.

### 3.5. Follow-Up Study

#### 3.5.1. GBM Cells with Knockdowns of the Selected Genes Demonstrate a Decreased Sensitivity to VSV-I and NDV-H2

The GBM cell line, DBTRG-05MG (ATCC^TM^), was chosen for knockdowns due to the following reasons: (1) DBTRG-05MG cells acquire protection against VSV-I after 24 h IFNα treatment before infecting [49]; (2) this cell line proliferates significantly faster in comparison with patient-derived cells, allowing achievement of culture establishing in adequate times. Since we aimed to identify conserved mechanisms, any GBM culture with proven intact IFNα signaling can be recruited to the follow-up. DBTRG-05MG sensitivity to VSV-I and enteroviruses was also quantitatively characterized by measuring the tissue culture infectious dose required to reach the fifty percent endpoint, lgTCID50%/mL, proving its preserved response to IFNα (Figure 4, “wt” row). To control off-target effects associated with the shRNA method, the cell line expressing hairpin to GFP2 was obtained (Figure 4, the “sh cntr” row). Compared with wild-type DBTRG-05MG cells, no statistically significant differences in cell sensitivity and virus replication were observed that confirmed the absence of undesirable effects.

We assumed that suppressing the key genes responsible for antiviral defense can increase the sensitivity of cells to viruses, including the oncolytic ones. However, the results were not straightforward enough (Figure 4). While MX1-deficient DBTRG-05MG cells demonstrated increased sensitivity to VSV-I and PV3, a knockdown of MX1 did not change cell sensitivity to other viruses tested. The IFN-treated cells with silenced OAS2, IFIT3, PLSCR1, and BST2 genes, have shown a statistically significant increase in the sensitivity to VSV-I, Coxsackievirus B5, and NDV-H2. The shRNA-mediated silencing of IFIT3, PLSCR1, and BST2 genes also reduced the sensitivity of DBTRG-05MG to VSV-I and NDV-H2 in the absence of IFN stimulation. The effect was also reproduced in normal astrocytes with silenced IFIT3 and PLSCR1 genes for VSV-I and PV3 (Appendix A). The observation of a statistically significant decrease in the sensitivity of untreated cells with the shRNA-mediated knockdowns was convincingly reproduced in quadruplicates using various viruses and cell cultures. In our experiments, a knockdown of a single interferon-stimulated gene, encoding a key regulator of the antiviral IFN response, could not confer increased sensitivity of untreated cells to tested viruses. Thus, we hypothesized that the suppression of OAS2, IFIT3, PLSCR1, and BST2 in our experiments could increase the overall expression of other ISGs in the cells and alter processes of virus entry into cells and/or their replication.

We measured the replication efficiency of viruses in a panel of DBTRG-05MG subcultures (Figure 4) to find if the decreased cell sensitivity can be caused by negative regulation of virus replication. In the cells with suppressed OAS2, BST2, IFIT3, and PLSCR1 and overexpressed PLSCR1, the VSV-I production was reduced compared to the parental DBTRG-05MG cells. These observations were partially reproduced for the other viruses indicating a negative effect of these gene manipulations on virus replication. For some of the data (untreated cells with suppressed BST2, IFIT3, and PLSCR1), the reduced VSV-I replication correlates with the reduced cell sensitivity to viruses, supporting the interrelation of these observations. The remaining data demonstrates a variety of effects requiring alternative hypotheses. For example, IFN-treated cells with suppressed MX1, OAS2, and BST2 compared to parent cells, revealed significantly increased NDV-H2 production without changing the cell sensitivity. Data in Figure 4 also show that the replication of viruses can go down upon overexpression or silencing of target genes; however, the cell sensitivity remains unchanged. There are examples of unchanged virus production at the decreased sensitivity of cells to viruses (e.g., IFN-untreated cells with silenced OAS2, IFIT3, and PLSCR1, infected with NDV-H2).

#### 3.5.2. Untreated DBTRG-05MG Cells with Suppressed IFIT3 and PLSCR1 Demonstrate an Increased Level of ISG Products and Changes in Endocytosis

We performed an additional proteomic analysis to determine quantitative changes in the DBTRG-05MG cells with downregulated IFIT3 and PLSCR1 genes (Figure 5, Appendix A). Besides identifying processes affected by the silencing of these genes, we aimed to assess the changes in the relative abundances of ISG protein products. Although the fold changes in baseline levels of these proteins were found not to be as dramatic as in response to IFN stimulation, they were statistically significant for about a hundred proteins. According to the Interferome database [48], the totality of these proteins exhibits a pattern characteristic of the products of IFN-stimulated genes. Thus, suppression of IFIT3 and PLSCR1 caused an increase in the abundances of APOL2, DDX58, ERAP1, HLA-B, HLA-C, GLS, MX1, ISG15, PML, PSME2, and SAMHD1 gene products. These observations agree with our expectations, although activating these genes alone is not enough to explain the observed resistance to viruses. Subsequent KEGG pathway analysis revealed an enrichment in endocytosis, protein processing in the endoplasmic reticulum, and MHC class I molecules (Appendix A). 

## 4. Discussion

Aiming to study the contribution of interferon mechanisms to the problem of personalized resistance of malignant glioblastoma to therapy with oncolytic viruses, we began by determining the response of short-term patient-derived GBM cell cultures to treatment with type I interferons. By comparing the portraits of ISGs with measurements of cell sensitivity to VSV-I and several oncolytic enteroviruses, we evaluated the quality of IFN-induced resistance acquired by GBM cells. We recorded changes at the level of mRNA and protein, changes in cell viability after infection, and infectious titers of produced viral particles.

We identified a conservative molecular pattern (genes listed in Figure 2a,b) that unambiguously differentiates between preserved and defective responses to type I interferon treatment. In cells with impaired IFN signaling, there were neither changes in the transcription of IFN-induced genes nor an increase in the levels of protein products involved in this biochemical process. This pattern was generally independent of the type I IFN subtypes used (IFNα and IFNβ). Our experiments confirm the previously established fact that IFNα elicits a significantly stronger antiviral response compared with IFNβ. In our experiments, we treated GBM cells with IFNα-2b and IFNβ at 100 and 1000 units/mL concentrations, respectively, to achieve comparable response amplitudes at the protein level. Differences in the activity of IFN isoforms are usually explained by differences in their affinity for the IFN receptor IFNAR [50,51,52]. When we compared GBM responses to IFNα and IFNβ, we did not observe meaningful and statistically significant differences (Appendix A, sheet titled “NSAF_IFNα_to_IFNβ_intact”). Probably, the proteomic coverage of 5000 quantitatively defined proteins is not deep enough to reveal this difference. Presumably, this difference does not exist since IFNα and IFNβ occupy the same receptor and act through the same JAK-STAT signaling pathway, initiating transcription of IFN-stimulated genes [52]. Therefore, we further focused our research on the characterization of the response to IFNα.

Using omics-derived ISGs portraits (Figure 2), measurements of the cell sensitivity to several viruses (Figure 1b), as well as the titers of the produced infectious viral particles, we evaluated the “strength” of IFN-induced resistance acquired by individual GBM cell cultures. First, to rank GBM cultures, we proposed a statistically valid “star” code (Figure 1b) and, according to the increase in response to IFN processing, ordered GBM cultures as follows: GBM4114, GBM5222, (GBM3821, GBM5410, GBM6067), GBM5522, GBM6138, and GBM4466. Then we noticed that the statistical evaluation of the omics data also shows similar relationships. Namely, lower *p*-values for fold changes in gene expression and protein regulation corresponded to a more robust response to IFN treatment (Appendix A, Appendix A). Thus, considering the statistical significance as a parameter for ranking the “strength” of the cellular response, transcriptomics allows one to arrange cultures in the following order: GBM4114, (GBM5522, GBM4446), GBM3821, while proteomics as GBM4114, GBM5522, GBM3821, and GBM6067 (Figure 3c). We see that the ranks determined by different methods are substantially similar, which instills confidence in these results. We also analyzed cross-associations between omics data and titration-based measurements (Appendix A, Appendix A). GO, KEGG, and Reactome analyses showed that genes and proteins, for which differential expression is correlated with differences in cell sensitivity to VSV-I and VSV-I replication (as high as abs(R) ≥ 0.75), are involved in biological processes associated with viral infection and replication, respectively.

Our pilot study deals with a very limited sample population, which restricts the ability to assess the accuracy and sensitivity of the approach used to rank IFN-induced resistance, and prevents complex data integration to generate new clinically relevant insights. In an effort to solve the global problem of rapid personalized assessment of the oncolytic potential of viruses to overcome IFN-induced resistance of tumor cells, we consider our study as the first step towards ranking glioblastoma cells in terms of their sensitivity to viruses. Our next goal would be to obtain a more representative set of GBM samples from patients, which will allow clinically relevant conclusions.

An attempt to detect the main interferon-stimulated genes mediating the sensitivity of glioblastoma multiforme cells to oncolytic viruses demonstrated that suppression of a single gene encoding the most important components of the antiviral response to type I interferons does not necessarily lead to an increase in the sensitivity of cells to oncolytic enteroviruses or to an increase in viral replication (Figure 4). Probably, in order to achieve such differences, the simultaneous suppression of several IFN-stimulated genes is required. In-depth studies are required to determine the specific contribution of individual IFN-stimulated genes to viral resistance. As an example, we can cite the recent work [53], which showed the role of the TRIM69 gene in the acquisition of resistance to VSV-I by cells.

Changes in the levels of individual IFN-stimulated genes can cause secondary changes in the expression of a number of genes, which themselves can affect the sensitivity of cells to viruses. Thus, we found that the shRNA-mediated suppression of the IFIT3 and PLSCR1 genes leads to an increase in the levels of ISG products and changes in gene products related to endocytosis. Subsequent analysis of the KEGG pathway revealed increased endocytosis, protein processing in the endoplasmic reticulum, and MHC class I molecules (Appendix A). If the knockdown negatively regulates the endocytosis pathway, it can prevent viruses from entering cells. For example, VSV-I enters cells through the clathrin-mediated endocytic pathway [54]. Thus, a decrease in VSV-I replication and a decrease in virus susceptibility in untreated cells may result from a lower permissiveness of cells along this pathway. In contrast, polioviruses [55] and Coxsackie A9 virus [56] use clathrin- and caveolin-independent pathways of entry into cells, while Echovirus1 can use dynamin-dependent [57] or dynamin-independent [58] endocytosis. The Newcastle disease virus can enter cells through caveola-mediated [59] and dynamin-dependent endocytosis [60]. The use of alternative routes of entry into cells may explain why suppression of OAS2, IFIT3, BST2, and PLSCR1 had little effect on cell sensitivity to other viruses and viral replication efficiency. However, we recognize the limitations of the shRNA-mediated knockdowns. Future independent knockdown and/or knockout experiments should be performed to obtain a more comprehensive picture of virus sensitivity and gene expression changes.

## 5. Conclusions

We conceived this project to study the contribution of interferon mechanisms to the defense of malignant cells against oncolytic viruses. Omics profiling of glioblastoma multiforme cultures allowed us to assess the diversity of molecular responses to type I IFNs. It has been demonstrated that the magnitudes of cellular responses at both protein and mRNA levels are in good agreement with the morphology- and titration-based estimations of cytopathic effects. This study has shown that the reaction of glioblastoma cells to interferon stimulation can be highly similar to that of normal cells and consists of interferon-stimulated genes and their proteins. Omics analysis detected end products of the interferon-induced cascade of biochemical reactions, allowing unambiguous differentiation between preserved and defective interferon signaling.

Titration-based characterization of malignant cells for sensitivity to a panel of viruses allowed classification of GBM cells by interferon-induced protection against viruses. Furthermore, it demonstrated that the protection of malignant cells is not solely based on the successful production of antiviral proteins. Instead, the ability of viruses to overcome the IFN-induced resistance would largely contribute to successful therapy.

Our attempt to discover essential interferon-stimulated genes mediating the glioblastoma multiforme sensitivity to oncolytic viruses demonstrated that suppression of MX1, OAS2, IFIT3, and PLSCR1, and overexpression of PLSCR1, encoding crucial components of antiviral response to type I interferons, does not necessarily result in acquiring an increased sensitivity to enteroviruses or enhanced virus replication. Silencing the targeted genes can increase the expression of other IFN-regulated counterparts and alter the endocytic pathways used for virus entry into the cells. However, this is a point for further validation using several shRNAs per gene knockdown and/or with the CRISPR-Cas9 approach to exclude off-target effects.

## Figures and Tables

**Figure 1 cancers-13-05268-f001:**
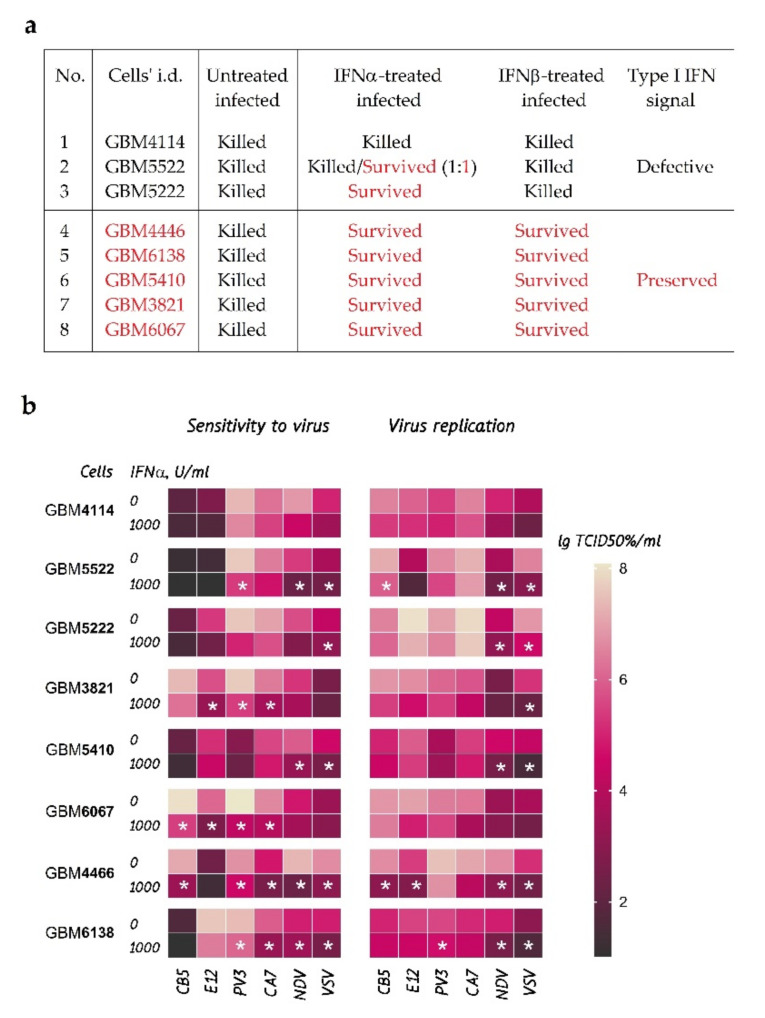
Cytopathic effects (CPEs) in eight GBM cell cultures, pretreated with IFNα or IFNβ, and untreated control. (**a**) CPEs of VSV-I by morphology. Test conditions for (**a**): pretreatment with 100 units/mL of IFNα-2b, or 1000 units/mL of IFNβ-2b for 24 h, infection with VSV-I at MOI = 1.0. CPEs were observed after 48 h. Culture GBM5522 showed CPE in 2 out of 4 repeats. (**b**) Cell protection against viruses (sensitivity) and virus replication was measured by virus titration. The highest lgTCID50/mL corresponds to the most profound decrease in glioblastoma cells and an increase in the number of virions. Statistically significant changes in sensitivity and replication are calculated using *t*-test corrected for multiple comparisons with the two-stage linear step-up procedure of Benjamini, Krieger, and Yekutieli (Q-value < 0.01), highlighted by “stars”. Abbreviations: VSV, Vesicular stomatitis virus Indiana strain; PV3, Sabin vaccine strain of type 3 poliovirus; NDV, Newcastle disease virus strain H2; CB5, Coxsackie B5 virus strain LEV14; CA7, Coxsackie A7 strain LEV8; E12, Echovirus 12 strain LEV7.

**Figure 2 cancers-13-05268-f002:**
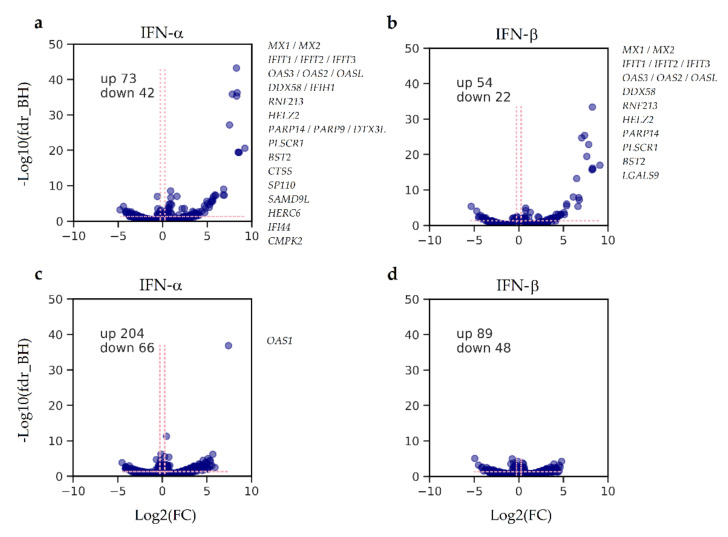
Alterations in GBM cells’ proteomes depending on the status of type I IFN signaling and IFN subtype used: (**a**,**b**) cohort with preserved antiviral protection by type I IFNs; (**c**,**d**) cohort with defective antiviral protection by type I IFNs. The genes named by plots correspond to the proteins altered as strong as log_2_(fold_change) > 5.0, with statistical significance as high as -log_10_(fdr_BH) > 5.0. Dashed lines correspond to statistical (fdr_BH < 0.05) and fold change (|log_2_FC| ≥ 0.263) thresholds used for the determination of up and downregulated proteins. NSAF used for missing value imputation MV = min(NSAFs)∙10^−3^.

**Figure 3 cancers-13-05268-f003:**
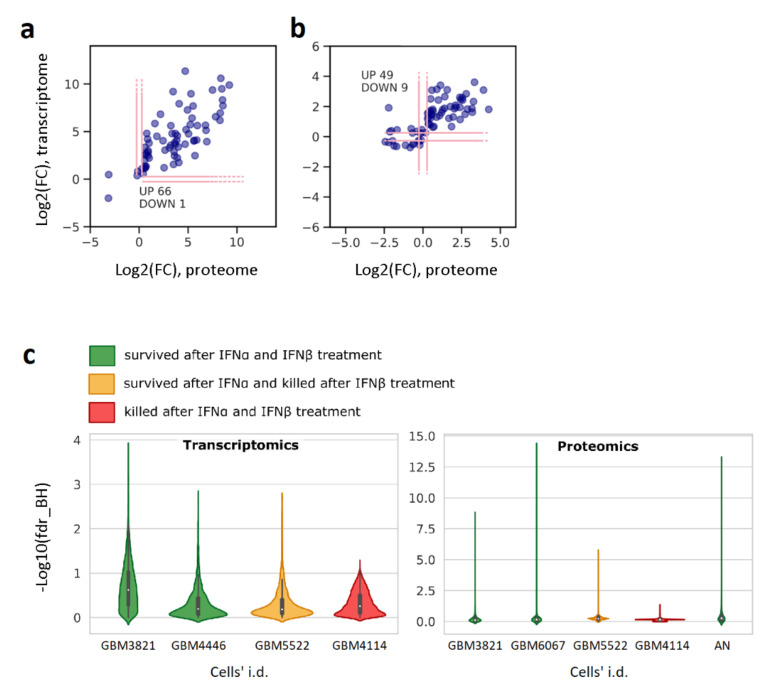
The mRNA and protein co-expression in IFNα-treated cells: (**a**) human skin fibroblasts [33] vs. GBM cultures protected from VSV-I by IFNα/IFNβ treatment (cultures marked as “preserved” in Figure 1a); (**b**) The IFN-protected GBM3821. Protein products and the respective genes satisfying fdr_BH < 0.05 were subject to analysis. (**c**) In terms of statistical significance, amplitudes of transcriptomic and proteomic cellular responses are correlated with CPEs summarized in Figure 1a. Abbreviation: AN, normal astrocytes.

**Figure 4 cancers-13-05268-f004:**
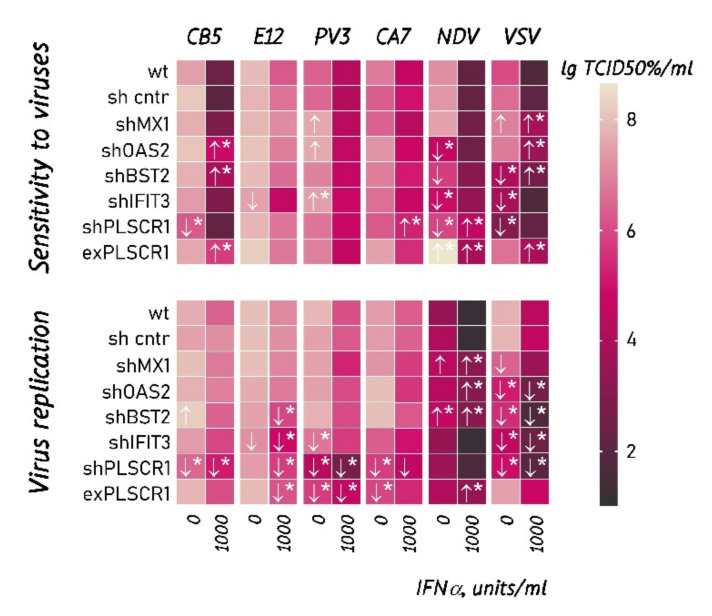
Sensitivity to viruses and virus replication of DBTRG-05MG and their derivatives with overexpression or shRNA-mediated silencing of genes for VSV-I and enteroviruses. The sensitivity and replication were measured in quadruplicates for a range of IFNα concentrations by virus titration in the sensitive RD cells. Cell cultures were treated with IFNα for 24 h, followed by infection with a virus. The tissue culture infectious dose required to reach the fifty percent endpoint, lgTCID50/mL, was measured 24 h after infection. Labels: ↑↓ denote a significant increase or decrease in cell sensitivity/virus replication with *t*-test *p*-value < 0.05, * significance corrected for multiple comparisons using the two-stage linear step-up procedure of Benjamini, Krieger, and Yekutieli, Q-value < 0.05 (GraphPad Prism). Abbreviations: VSV, Vesicular stomatitis virus Indiana strain; PV3, Sabin vaccine strain of type 3 poliovirus; NDV, Newcastle disease virus strain H2; CB5, Coxsackie B5 virus strain LEV14; CA7, Coxsackie A7 strain LEV8; E12, Echovirus 12 strain LEV7.

**Figure 5 cancers-13-05268-f005:**
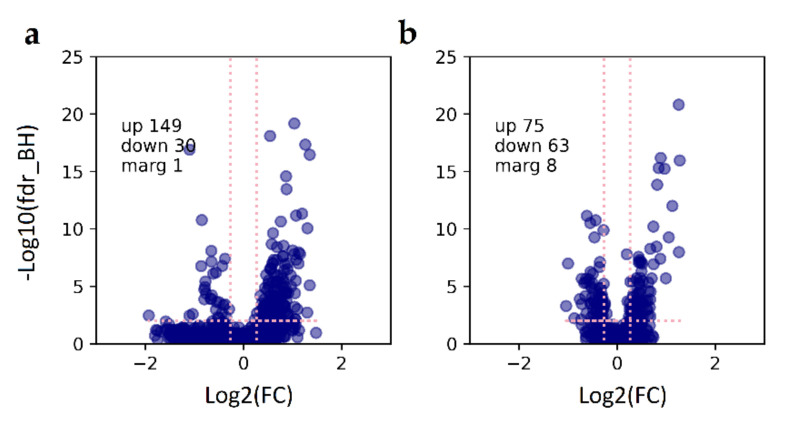
Comparison of DBTRG-05MG cells with silenced IFIT3 (**a**) and PLSCR1 (**b**) with the parental DBTRG-05MG cells at the proteome level. Differentially regulated proteins satisfy fdr_BH < 0.01 and FC ≥ 1.2. Statistical significance was calculated with Diffacto [43] adapted for use with TMT11plex experiments.

## Data Availability

Proteomics data are deposited to ProteomeXchange (projects PXD022836 and PXD022906). The RNA sequencing data is available at NCBI Gene Expression Omnibus (GEO) under the accession number GSE163949.

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
