# Peer review of "Multi-Omics Analysis of Glioblastoma Cells’ Sensitivity to Oncolytic Viruses"

_cancers, 2021, doi:10.3390/cancers13215268_

Round 1

Reviewer 1 Report

The authors studied the molecular basis of interferon to the defense of malignant cells against oncolytic viruses. Overall, experiments are well designed, and the manuscript is well written but still needs to be properly balanced with caveats and uncertainties noted.  I have a few concerns, which authors should address before publication:

  1. It looks like some “heat colors” are more significant than other “stars”, please include the raw data in the supplementary table and clarify the details of statistical analysis in the methods section.
  2. It is obvious that the microphotographs of cells with different treatments are the same in Figure 2a, and a similar problem happened in Figure 2b. In Figure S4A, the same images were used to represent three kinds of cells (astrocytes, GBM6067, and 3821). Magnification should be indicated in the microphotographs.
  3. Considering profound off-target effects of siRNA-mediated knockdown, rescue experiments using the siRNA-resistant forms of antiviral proteins must be performed. mRNA levels also need to be evaluated via Q-PCR to confirm the RNA knockdown. Because shutdown of gene expression using CRISPR-Cas9 systems can easily be done these days, the authors should confirm some of their results using CRISPRi in Figures 4 and 5.

Author Response

Dear Reviewer,

We are sending the revised manuscript by Lipatova et al., "Multi-Omics analysis of glioblastoma cells' sensitivity to oncolytic viruses," which was modified to meet the criticism and suggestions made during the evaluation process. We thank for valuable and constructive suggestions, which helped us significantly improve the revised manuscript. Below is the point-by-point description of the modifications.

Reviewer #1

Comments and Suggestions for Authors

The authors studied the molecular basis of interferon to the defense of malignant cells against oncolytic viruses. Overall, experiments are well designed, and the manuscript is well written but still needs to be properly balanced with caveats and uncertainties noted. I have a few concerns, which authors should address before publication:

  1. It looks like some "heat colors" are more significant than other "stars", please include the raw data in the supplementary table and clarify the details of statistical analysis in the methods section.

Author's reply: Agree. We included raw data with the details of statistical analysis in the supporting information (Table S1) and methods section (p. 5 of the revised manuscript). Statistical evaluation of data was performed using GraphPad Prism 8.0.2 software using the following settings: Unpaired t-test, correction for multiple comparisons using Two-stage step-up (Benjamini, Krieger, and Yekutieli) method, Desired FDR (Q-value) (for Fig. 1b) < 1.00%, Desired FDR (Q-value) (for Fig. 4) < 5.00%. Satisfying the FDR condition was marked by "stars." Note that if standard deviations for the compared means are large and overlapped, testing will not result in statistically significant discoveries. That is the reason why visually significant differences in "heat colors" can lack "stars."

  1. It is obvious that the microphotographs of cells with different treatments are the same in Figure 2a, and a similar problem happened in Figure 2b. In Figure S4A, the same images were used to represent three kinds of cells (astrocytes, GBM6067, and 3821). Magnification should be indicated in the microphotographs.

Author's reply: Microphotographs of VSV-I -resistant (DBTRG-05MG cells in Figure 2 and S4A) and VSV-I -destroyed (A172 in Figure 2 and S4A) cells were done by us in our earlier work [1]. In the current study, we used the photos of alive and dead cells to mark the samples if they have preserved or defective antiviral responses, and not for any other purpose. Since it causes substantial misinterpretation, we revised Figure 2 and Figure S4a and removed microphotographs.

1. Tarasova, I.A.; Tereshkova, A.V.; Lobas, A.A.; Solovyeva, E.M.; Sidorenko, A.S.; Gorshkov, V.; Kjeldsen, F.; Bubis, J.A.; Ivanov, M.V.; Ilina, I.Y.; et al. Comparative proteomics as a tool for identifying specific alterations within interferon response pathways in human glioblastoma multiforme cells. Oncotarget 2018, 9, 1785-1802, doi:10.18632/oncotarget.22751.

  1. Considering profound off-target effects of siRNA-mediated knockdown, rescue experiments using the siRNA-resistant forms of antiviral proteins must be performed. mRNA levels also need to be evaluated via Q-PCR to confirm the RNA knockdown. Because shutdown of gene expression using CRISPR-Cas9 systems can easily be done these days, the authors should confirm some of their results using CRISPRi in Figures 4 and 5.

Author's reply: We agree that CRISPR-Cas9 presents an alternative to shRNA-mediated knockdowns performed in our study. However, we opted for the latter approach. We favor the shRNA approach because, unfortunately, the CRISPR-Cas9 technology is also prone to off-target effects [doi:10.1038/nbt.2623, doi:10.1101/gr.162339.113]. Also, during the timely process of knockout cell selection in the CRISPR-Cas9 method, there could be additional secondary changes in the expression of some genes. Therefore, in the study, we used freshly obtained mass cultures of shRNA-mediated knockdown cells that represent many independent gene silencing events. By obtaining a mass cell culture from fresh tissues and testing it in a short time, we avoided secondary effects associated with obtaining individual clones and their long-term cultivation, which inevitably lead to significant changes in the cell phenotype, first of all in its sensitivity to viruses. This is crucial, since we must keep cells most close to that phenotype, which was observed from patients. Otherwise, efficiency of viral replication and sensitivity to oncolytic viruses of cell clones change and do not reproduce that parameters in the primary cells. Thus, we lose the predictive feasibility for patient sensitivity to oncolytic virus. Authors have observed such effects after CRISPR-Cas9 knockout [DOI: 10.31857/S0026898420040102], therefore the shRNA approach, which does not impose long cultivation times, was deliberately chosen.

To estimate the potential contribution of the off-target effects associated with the shRNA method, a corresponding line expressing a hairpin to GFP2 was obtained (Figure 4 of the revised manuscript, the “sh ctrl” row). Compared with the wild type DBTRG-05MG cells, no statistically significant differences for cell sensitivity and virus replication were observed that confirm the absence of undesirable effects of the shRNA method.

To further address the Reviewer's concern, we supplemented results of real-time qPCR to confirm knockdowns at mRNA level in Figure S1 (Supporting Information). The methods section on p. 6 was revised accordingly.

Reviewer 2 Report

The described work aims to investigate a patient-specific response of glioblastoma cells to infection by oncolytic viruses in an interferon-dependent manner.

The authors use patient-derived glioblastoma cell lines infected with different oncolytic viruses and assay patient-specific cytopathic effects, effects of different interferons on protein expression, correlations of proteome and transcriptome, and the effect of knockdown/overexpression of genes with presumed antiviral activity on sensitivity to virus/replication.

Major concerns:

  1. A lot of experimental work has gone into this study but the scientific conclusions are quite limited. The authors rightly state that suppression of the selected genes does not conclusively increase virus sensitivity. It is questionable if this is due to some experimental concerns or down to the selected genes, which were a combination of known antiviral genes and those with unclear role. Further information on the experimental setup (numbers of technical and biological replicates, information on the knockdown quality on mRNA level) as well as a clearer rationale on how the target genes were selected could be helpful.
  2. The work is described as a multi-omics analysis approach, however, the bioinformatical approaches although of state-of-the -art quality on the single omics level, lack some creativity on multi-modal integration. For inspiration I suggest (amongst many others): Subramanian I, Verma S, Kumar S, Jere A, Anamika K. Multi-omics Data Integration, Interpretation, and Its Application. Bioinform Biol Insights. 2020;14:1177932219899051. Published 2020 Jan 31. doi:10.1177/1177932219899051
  3. The whole study is hampered by the availability of only 8 patient samples. An increase of study subjects would have drastically increased the scientific merit and impact.

Author Response

Dear Reviewer,

We are sending the revised manuscript by Lipatova et al., "Multi-Omics analysis of glioblastoma cells' sensitivity to oncolytic viruses," which was modified to meet the criticism and suggestions made during the evaluation process. We thank for valuable and constructive suggestions, which helped us significantly improve the revised manuscript. Below is the point-by-point description of the modifications.

REVIEWER #2

Comments and Suggestions for Authors

The described work aims to investigate a patient-specific response of glioblastoma cells to infection by oncolytic viruses in an interferon-dependent manner.

The authors use patient-derived glioblastoma cell lines infected with different oncolytic viruses and assay patient-specific cytopathic effects, effects of different interferons on protein expression, correlations of proteome and transcriptome, and the effect of knockdown/overexpression of genes with presumed antiviral activity on sensitivity to virus/replication.

Major concerns:

  1. A lot of experimental work has gone into this study but the scientific conclusions are quite limited. The authors rightly state that suppression of the selected genes does not conclusively increase virus sensitivity. It is questionable if this is due to some experimental concerns or down to the selected genes, which were a combination of known antiviral genes and those with unclear role. Further information on the experimental setup (numbers of technical and biological replicates, information on the knockdown quality on mRNA level) as well as a clearer rationale on how the target genes were selected could be helpful.

Author's reply: As suggested, we added information on biological replicates, details, and results of statistical analysis for titration-based measurements of cell sensitivity to viruses and virus replication (Methods section, p.5 of revised manuscript) and Table S1 (Supporting Information). Also, we supplemented results with qPCR confirming knockdowns at mRNA level (Figure S1, Supporting Information) and revised text in the Methods section (p.6) accordingly.

To further address the Reviewer's concern, we added more details about selecting the target genes on page 14 of the revised manuscript. Briefly, the logic was as follows: Based on the omics data, we hypothesized that proteins exhibiting the most profound IFN-induced quantitative changes in GBM cells could be among the products of genes that usually determine the resistance of normal cells to viruses in response to treatment with Type I IFNs. To choose the target genes, we used results of hierarchic clustering of quantitative changes (with FC > 2.0) across the samples with preserved IFN-induced antiviral protection (Figure S5), which integrates "averaged" (Figure 2) and "personalized" (Figure S4) cell responses. Genes encoding proteins identified across at least four out of five sample comparisons were considered as promising targets. Such a "core" response across the sample comparisons counted about 30 proteins, which we empirically split into two subgroups: (1) well-studied proteins with antiviral activity (MX1/2, IFIT1/2/3, OAS1/2/3/L, etc.), and (2) proteins with understudied roles in response to IFNs (PLSCR1, BST2, PARP9, etc.). Of each subgroups, we randomly selected a few genes: (1) well-studied (IFIT3, OAS2, and MX1), and (2) understudied (PLSCR1 and BST2). The goal was to identify genes whose suppression or overexpression affects GBM cells' sensitivity to a panel of potentially oncolytic viruses (NDV-H2, PV3, Echo12, Coxsackie A7, and Coxsackie B5) used in this study.

  1. The work is described as a multi-omics analysis approach, however, the bioinformatical approaches although of state-of-the -art quality on the single omics level, lack some creativity on multi-modal integration. For inspiration I suggest (amongst many others): Subramanian I, Verma S, Kumar S, Jere A, Anamika K. Multi-omics Data Integration, Interpretation, and Its Application. Bioinform Biol Insights. 2020;14:1177932219899051. Published 2020 Jan 31. doi:10.1177/1177932219899051

Author's reply: Yes, we separately evaluated each block of data (proteomics, transcriptomics, and titration-based measurements) followed by analyzing the co-directed (co-ranked) properties. However, it was our deliberate choice because the sample populations and their overlapping in our pilot study are pretty limited (as was correctly mentioned by the Reviewer in the following comment). Furthermore, since the primary goal of multi-omics data integration is biomarker discovery, disease subtyping/classification, and deriving insights into disease biology based on the analysis of hundreds of samples (as described in the review suggested by the Reviewer), using those tools in our study would not produce sound results. However, we applied efforts to show that analysis of titration-based measurements of cell sensitivity, virus replication together with the omics data demonstrates rank coincidence for the status of type 1 interferon signaling in GBM cells, determined by different methods (Discussion section on p.17, Figure S10, Table S7 of the revised Supporting Information). That was our primary motivation behind the study.

  1. The whole study is hampered by the availability of only 8 patient samples. An increase of study subjects would have drastically increased the scientific merit and impact.

Author's reply: We agree that the sample population in the current study is too limited to achieve significance in terms of clinical applications. Aiming at the global task of fast monitoring the oncolytic potential of viruses able to overcome IFN-induced resistance of tumor cells, we have started from a pilot study addressing why patient-derived glioblastoma cultures can acquire increased resistance to oncolytic viruses in the presence of interferons and suggesting an approach to ranking glioblastoma cells by the acquired resistance. We elaborated on this issue in the Discussion section (p.17 of the revised manuscript). Collecting a more representative population of GBM samples from patients approaching clinical levels to extend further the present study is currently underway.

Reviewer 3 Report

Lipatova et al. have performed an extensive analysis on the role of IFN role in sensitivity to oncolytic viruses. This work not only includes multi-omics analysis, since the results are further supported with cell biology experiments. Altogether is a very interesting manuscript that sometimes lack a bit of clarity, mostly due to the inherent complexity of the experiments. Here, some major and minor concerns, that include some suggestions.

Major concerns:

- In guidelines for authors: “Research articles have a standard structure, which is set out in the instructions for authors of the journal and the journal template. The majority of journals use a so-called IMRAD structure, meaning the sections are Introduction, Materials and Methods, Results, and Discussions.” In cancers the template indicates the IMRAD structure.

-line-305 When the authors indicate that “treatment with IFNα or IFNβ successfully protected cells from VSV-I-induced death”, it is not clear the type of experiments that they have conducted to reach this conclusion. In the M&M section it is described “Then, IFN-treated and untreated (control) cells were infected with VSV-I (Indiana strain) at the wide range multiplicity of infection (MOI) from 100 to 0.0001. Cytopathic effects (tissue culture infection dose, lgTCID 50/mL) were estimated after 24 and 72 h following the infection.” Finally, in the figure legend, we can read “by morphology,”. The way how the authors reach the conclusions of fig1a, are not properly described in the main text. No representative pictures are been shown and the criteria to consider them defective/preserved is not clear. If there are any quantification, this should be displayed too.

-line 320, The authors consider” Statistically significant IFN-induced changes in cell sensitivity and virus replication for each culture are highlighted by “stars””; however, some of these changes are 25% decrease and others even 50%. This fact is not reflected in the star code, and may influence the final ranking of sensitivity.

- In order to be more informative, figure 2 should be label as a, b, c, d and those labels should be indicated in Figure S2. If the at the end of the design, the boxes are related to Fig 1c, Fig 1d, Fig 1a, Fig 1b (according their actual order in the schema), that would help to understand the experiments behind the analysis. This labeling may also help to understand the main text where it is not clear which comparison are referring to in some cases. Same for fig S3 and S4.

- Authors should comment how in “defective “group, having more statistically significant protein changed, only one pass the cutoff imposed.

- Since the total list of proteins is not in the figure 2, the analysis “most of the altered proteins corresponded to known IFN-responsive antiviral proteins, as well as the proteins involved in the innate immune response (according to the Interferome database [48] and GO analysis using STRING [49).” Should be shown somewhere, preferably as a chart in the figure.

- The picture in the right corner of volcano plots is very confusing. It is shown as a way to mark preserved vs. defective GBM lines. However, they are the same picture independently of the cell line. it could have been considered as a representative of the pool of samples, but volcanos in S4 corresponds just to one cell line. In the figure legend S4 these microphotographs are not even mentioned.

-“Comparing data from the virus susceptibility tests and the proteome analyses using different statistical designs suggests that with more robust antiviral protection, higher fold change values and lower p-values are observed for protein products of IFN-responsive genes. Notably, all upregulated proteins were recognized as IFN-induced across the public ISG databases [33-48]. GO analysis was performed to ensure enrichment of type I IFN signaling pathways. The GBM cultures protected from VSV-I by IFN treatment demonstrated enrichment on the upregulated proteins of the antiviral response and interferon signaling, including the JAK-STAT cascade.”. This data that they are discussing it is not shown in the figures and there is no reference to a Table.

-It is not clear why the authors perform the analysis in 3.3 “after the treatment with IFNα in the IFN-protected (“preserved”) group of GBM cultures” but not after the treatment with IFNβ.

- Section 3.5. maybe not well designed. First, the authors use DBTRG-05MG as original cell line to knockdown, but we don’t know the sensitivity profile of this wt line, because all the analyses have been done in patient derived GBM cell lines. Second, the knockdown of each gene has not been properly quantified by qPCR (RNA) and maybe that WB (protein) doesn’t properly reflect the real silencing. Third, the authors have not considered the possibility that these genes may interact among then and therefore single knockdown may be not that effective.

Minor concerns

-It would be easier to follow if the patient’s samples name were not just a number, maybe #number, or GBM-CL (GBM cell line) +number (It’s up to the authors). In this way we could distinguish their name from other numbers.

-Numbers of the figures in the left-up corner, as they do in other papers, would be clearer. No need to put them in brackets in the figure (but keep it in the figure legend)

-The virus tested in Fig 1B, should be explain at least in the figure. (meaning of the abbreviations).

-In figure 1. If “Test conditions for (A): pre-treatment with 100 units/mL of IFNα-2b, or 1000 units/mL of 334 IFNβ-2b for 24 hours, infection with VSV-I at MOI = 1.0. CPEs were observed after 48 hours. Culture 5522 showed CPE in 335 2 out of 4 repeats.” Corresponds to (a), then it is clearer if presented after the explanation of (a) and before mentioning (b).

- In order to follow the paper “defective” group, should be referred as such throughout the paper. (i.e.line378)

-To follow the same way of presenting suppl. data. In line 340: “(S- Figures 2-3).” Should be “(Figures S2-3).”

-Line 375 “we observed a regular co-expression of about 30 proteins.” These proteins should be referred. If they are no listed anywhere, a Suppl table should be created.

Author Response

Dear Reviewer,

We are sending the revised manuscript by Lipatova et al., "Multi-Omics analysis of glioblastoma cells' sensitivity to oncolytic viruses," which was modified to meet the criticism and suggestions made during the evaluation process. We thank for valuable and constructive suggestions, which helped us significantly improve the revised manuscript. Below is the point-by-point description of the modifications.

REVIEWER #3

Comments and Suggestions for Authors

Lipatova et al. have performed an extensive analysis of IFN role in sensitivity to oncolytic viruses. This work not only includes multi-omics analysis, since the results are further supported with cell biology experiments. Altogether is a very interesting manuscript that sometimes lack a bit of clarity, mostly due to the inherent complexity of the experiments. Here, some major and minor concerns, that include some suggestions.

Major concerns:

- In guidelines for authors: "Research articles have a standard structure, which is set out in the instructions for authors of the journal and the journal template. The majority of journals use a so-called IMRAD structure, meaning the sections are Introduction, Materials and Methods, Results, and Discussions." In cancers the template indicates the IMRAD structure.

Author's reply: Agree. We revised the manuscript and added Discussions as a separate section to satisfy journal guidelines (p. 16-18).

-line-305 When the authors indicate that "treatment with IFNα or IFNβ successfully protected cells from VSV-I-induced death," it is not clear the type of experiments that they have conducted to reach this conclusion. In the M&M section it is described "Then, IFN-treated and untreated (control) cells were infected with VSV-I (Indiana strain) at the wide range multiplicity of infection (MOI) from 100 to 0.0001. Cytopathic effects (tissue culture infection dose, lgTCID 50/mL) were estimated after 24 and 72 h following the infection." Finally, in the figure legend, we can read "by morphology,". The way how the authors reach the conclusions of fig1a, are not properly described in the main text. No representative pictures are been shown and the criteria to consider them defective/preserved is not clear. If there are any quantification, this should be displayed too.

Author's reply: In our work, we used two types of virus bioassays to determine if the GBM cells are protected against viruses by IFN treatment or not. The first test is a co-called "VSV assay," a standard virus assay, in which untreated control and IFN-treated cells (a range of IFN concentrations are usually considered) are infected with VSV at MOI = 1.0 [1,2]. Cytopathic effects (CPE) are estimated by cell morphology 24 h and/or 48 h after VSV infection. Typically, homogeneous cell culture unprotected against VSV is fully killed (visually, only single cells of 5000 cell populations survive) in 24 h, even if treated with high IFN doses (1000 units / mL). The cellular response is binary for the VSV test: either cells survived or were killed (Figure 1a). The second assay is quantitative, a standard titration-based approach for calculating 50% endpoints, suggested by Reed and Muench [3]. As recommended by the Reviewer, we revised the Methods section to clarify the criteria and relationships between data, methods, and the results (p.4-5).

1. Shaw, A.E.; Hughes, J.; Gu, Q.; Behdenna, A.; Singer, J.B.; Dennis, T.; Orton, R.J.; Varela, M.; Gifford, R.J.; Wilson, S.J.; et al. Fundamental properties of the mammalian innate immune system revealed by multispecies comparison of type I interferon responses. PLoS Biol 2017, 15, e2004086, doi:10.1371/journal.pbio.2004086.

2. Tarasova, I.A.; Tereshkova, A.V.; Lobas, A.A.; Solovyeva, E.M.; Sidorenko, A.S.; Gorshkov, V.; Kjeldsen, F.; Bubis, J.A.; Ivanov, M.V.; Ilina, I.Y.; et al. Comparative proteomics as a tool for identifying specific alterations within interferon response pathways in human glioblastoma multiforme cells. Oncotarget 2018, 9, 1785-1802, doi:10.18632/oncotarget.22751.

3. Reed, L.J.; Muench, H. A simple method of estimating fifty per cent endpoints The American Journal of Hygiene 1938, 27, 493-497.

-line 320, The authors consider" Statistically significant IFN-induced changes in cell sensitivity and virus replication for each culture are highlighted by "stars""; however, some of these changes are 25% decrease and others even 50%. This fact is not reflected in the star code, and may influence the final ranking of sensitivity.

Author's reply: Yes, that is true. "Star" code allows a rough rank of cultures. If many cultures have the same number of stars, we further can suggest ranking them based on the averaging statistical significance of "stars". Other scoring schemes were also tried (Table S1, “primary_cultures_statistics” sheet). Tested metrics account for mean differences and standard deviations and rank cultures in the same order. Thus, it should be more comprehensive. For example, cultures GBM3821, GBM5410, and GBM6067 have four stars each, their mean q-values are 0.0053, 0.001912, and 0.000767, respectively (Table S1, “primary_cultures_statistics” sheet). Thus, their ordering from weakest to strongest IFN response: GBM3821, GBM5410, and GBM6067. We revised the manuscript on p. 9 and added Table S1 (Supporting Information) to clarify the issue and address the Reviewer's concern.

- To be more informative, figure 2 should be labeled as a, b, c, d, and those labels should be indicated in Figure S2. If the at the end of the design, the boxes are related to Fig 1c, Fig 1d, Fig 1a, Fig 1b (according to their actual order in the schema), that would help to understand the experiments behind the analysis. This labeling may also help to understand the main text where it is not clear which comparison are referring to in some cases. Same for fig S3 and S4.

Author's reply: As suggested, we revised labeling in Figures 2, S2, S3, and S4 to clarify the statistical design underlying quantitative results.

- Authors should comment how in the "defective "group, having more statistically significant protein changed, only one pass the cutoff imposed.

Author's reply: The concern is most probably related to Figure 2. The only OAS1 was highlighted for the "defective" cohort of IFNa-treated GBM cultures, while many more proteins have passed fdr_BH < 0.05. The genes named by plots correspond to the proteins altered as strong as log2(fold_change) > 5.0, with statistical significance as high as -log10(fdr_BH) > 5.0. We revised the legend to Figure 2 in the revised manuscript to clarify this point.

- Since the total list of proteins is not in the figure 2, the analysis "most of the altered proteins corresponded to known IFN-responsive antiviral proteins, as well as the proteins involved in the innate immune response (according to the Interferome database [48] and GO analysis using STRING [49)." Should be shown somewhere, preferably as a chart in the figure.

Author's reply: To address this concern, we supplied the results of GO enrichment analysis in Table S5 (in the revised Supporting Information). The total list of all quantified proteins behind Figure 2 is provided in Table S4.

- The picture in the right corner of the volcano plots is very confusing. It is shown as a way to mark preserved vs. defective GBM lines. However, they are the same picture independently of the cell line. it could have been considered as a representative of the pool of samples, but volcanos in S4 corresponds just to one cell line. In the figure legend S4 these microphotographs are not even mentioned.

Author's reply: We agree. We revised Figure 2 and Figure S4 and removed the confusing pictures.

-"Comparing data from the virus susceptibility tests and the proteome analyses using different statistical designs suggests that with more robust antiviral protection, higher fold change values and lower p-values are observed for protein products of IFN-responsive genes. Notably, all upregulated proteins were recognized as IFN-induced across the public ISG databases [33-48]. GO analysis was performed to ensure enrichment of type I IFN signaling pathways. The GBM cultures protected from VSV-I by IFN treatment demonstrated enrichment on the upregulated proteins of the antiviral response and interferon signaling, including the JAK-STAT cascade.". This data that they are discussing it is not shown in the figures and there is no reference to a Table.

Author's reply: We supported the discussed data with references to Figures and Tables in the manuscript's main text and Supporting Information (p.11 of the revised manuscript).

-It is not clear why the authors perform the analysis in 3.3 "after the treatment with IFNα in the IFN-protected ("preserved") group of GBM cultures" but not after the treatment with IFNβ.

Author's reply: To address the comment, we discussed this question in the Discussion section (p.17). There were several reasons behind our choice:

1. Response to IFNa is much stronger compared with IFNb. To obtain comparable response amplitudes at the protein level, we treated GBM cells with IFNα-2b and IFNβ at 100 and 1000 units/mL concentrations, respectively (Methods section, p.4 of the revised manuscript). The difference in response to IFNa and IFNb stimulations as described elsewhere [1,2] and our data confirm this fact.

2. When we compared GBM responses to IFNa and IFNb, we did not observe meaningful and statistically significant differences (Table S4 of the revised Supporting Information). Probably, proteome coverage of 5000 quantified proteins is not deep enough to dig out the difference.

3. This difference probably does not exist since IFNa and IFNb occupy the same IFNAR receptor and act through the same JAK-STAT pathways to initiate the transcription of IFN-stimulated genes [3].

Therefore, we focused on searching public omics data characterizing a core response to IFNa in humans, paying the most considerable attention to statistical design of the study and methods used for proving the intact status of IFN signaling in the cells (particularly, VSV assay supported with either quantitative proteomics or transcriptomics at the molecular level). The best match was the paper by Shaw et al. [4].

1. Piehler, J.; Schreiber, G. Mutational and structural analysis of the binding interface between type I interferons and their receptor Ifnar2. J Mol Biol 1999, 294, 223-237, doi:10.1006/jmbi.1999.3230.

2. Runkel, L.; Pfeffer, L.; Lewerenz, M.; Monneron, D.; Yang, C.H.; Murti, A.; Pellegrini, S.; Goelz, S.; Uzé, G.; Mogensen, K. Differences in activity between alpha and beta type I interferons explored by mutational analysis. J Biol Chem 1998, 273, 8003-8008, doi:10.1074/jbc.273.14.8003.

3. Stark, G.R. How cells respond to interferons revisited: from early history to current complexity. Cytokine Growth Factor Rev 2007, 18, 419-423, doi:10.1016/j.cytogfr.2007.06.013.

4. Shaw, A.E.; Hughes, J.; Gu, Q.; Behdenna, A.; Singer, J.B.; Dennis, T.; Orton, R.J.; Varela, M.; Gifford, R.J.; Wilson, S.J.; et al. Fundamental properties of the mammalian innate immune system revealed by multispecies comparison of type I interferon responses. PLoS Biol 2017, 15, e2004086, doi:10.1371/journal.pbio.2004086.

- Section 3.5. maybe not well designed. First, the authors use DBTRG-05MG as original cell line to knockdown, but we don't know the sensitivity profile of this wt line, because all the analyses have been done in patient derived GBM cell lines. Second, the knockdown of each gene has not been properly quantified by qPCR (RNA) and maybe that WB (protein) doesn't properly reflect the real silencing. Third, the authors have not considered the possibility that these genes may interact among them and therefore single knockdown may be not that effective.

Author's reply: We agree that this section required revision to clarify all details of the experimental design. We revised Section 3.5 on p.14. First, DBTRG-05MG is the GBM cell line (ATCC), for which the sensitivity to IFNa was characterized using the VSV assay in our previous work [1]. It was demonstrated that DBTRG-05MG cells acquire resistance to vesicular stomatitis virus (VSV) after 24 h IFNa treatment before VSV infecting. Also, the present study quantitatively characterized DBTRG-05MG sensitivity to VSV and enteroviruses by measuring lgTCID50% /mL (Figure 4 of the revised manuscript, row titled as "wt"). Second, we chose this commercial line because it can propagate significantly faster than with patient-derived cells we have at hand. Since we aimed to identify genes whose expression affects the GBM sensitivity to viruses and considered these genes as participating in conserved mechanisms, any GBM culture with proven intact IFN signaling can be recruited to the follow-up.

We supplemented the results of mRNA measurements in DBTRG-05MG and its derivatives with gene knockdowns in Figure S1a, proving the silencing of targeted genes.

We agree that simultaneous silencing of several IFN-stimulated genes can serve as a more effective strategy. There are examples of it [2]. We added this viewpoint to the Discussion section (p.18 of the revised manuscript).

1. Tarasova, I.A.; Tereshkova, A.V.; Lobas, A.A.; Solovyeva, E.M.; Sidorenko, A.S.; Gorshkov, V.; Kjeldsen, F.; Bubis, J.A.; Ivanov, M.V.; Ilina, I.Y.; et al. Comparative proteomics as a tool for identifying specific alterations within interferon response pathways in human glioblastoma multiforme cells. Oncotarget 2018, 9, 1785-1802, doi:10.18632/oncotarget.22751.

2. Kueck, T.; Bloyet, L.M.; Cassella, E.; Zang, T.; Schmidt, F.; Brusic, V.; Tekes, G.; Pornillos, O.; Whelan, S.P.J.; Bieniasz, P.D. Vesicular Stomatitis Virus Transcription Is Inhibited by TRIM69 in the Interferon-Induced Antiviral State. J Virol 2019, 93, doi:10.1128/jvi.01372-19.

Minor concerns

-It would be easier to follow if the patient's samples name were not just a number, maybe #number, or GBM-CL (GBM cell line) +number (It's up to the authors). In this way, we could distinguish their name from other numbers.

Author's reply: Done, the format is GBM+i.d.

-Numbers of the figures in the left-up corner, as they do in other papers, would be clearer. No need to put them in brackets in the figure (but keep it in the figure legend)

Author's reply: We revised figures to meet format guidelines.

-The virus tested in Fig 1B, should be explain at least in the figure. (meaning of the abbreviations).

Author's reply: Done. We revised legends for Figure 1 and Figure 4 and added meanings of abbreviations.

-In figure 1. If “Test conditions for (A): pre-treatment with 100 units/mL of IFNα-2b, or 1000 units/mL of 334 IFNβ-2b for 24 hours, infection with VSV-I at MOI = 1.0. CPEs were observed after 48 hours. Culture 5522 showed CPE in 335 2 out of 4 repeats." Corresponds to (a), then it is clearer if presented after the explanation of (a) and before mentioning (b).

Author's reply: Done.

- To follow the paper "defective" group, should be referred as such throughout the paper. (i.e.line378)

Author's reply: Done.

-To follow the same way of presenting suppl. data. In line 340: "(S- Figures 2-3)." Should be "(Figures S2-3)."

Author's reply: Done.

-Line 375 "we observed a regular co-expression of about 30 proteins." These proteins should be referred. If they are no listed anywhere, a Suppl table should be created.

Author's reply: These proteins are all named in Figure S5 (Supporting Information), demonstrating results of hierarchic clusterization of the IFN-induced fold changes for upregulated proteins (p.11 of the revised manuscript).

Reviewer 4 Report

This is an interesting study aiming at establishing Omic - based criteria for the efficient use of oncolytic viruses to  treat Glioblastoma.  The concept is ambitious, the group seems to be highly experienced in the field and the approach is technically sound and efficient.  Because of the highly complex and /or antagonistic nature  of IFN response repertoire,   the authors were not able to  establish  a protein/gene profile that predicts virus response and oncolytic activity.   Although the results cannot be directly translated to clinical guidance , this  is a promising approach for further refinement. 

Few grammar/syntax comments are highlighted in the draft .  

In future studies, group might choose to employ CRISR-Cas genetic screens for the unbiased detection of   virus vulnerability genes.

Author Response

Dear Reviewer,

Thank you very much for your positive response. Our reply and description of revisions is below.

REVIEWER #4

Comments and Suggestions for Authors

This is an interesting study aiming at establishing Omic - based criteria for the efficient use of oncolytic viruses to treat Glioblastoma. The concept is ambitious, the group seems to be highly experienced in the field and the approach is technically sound and efficient. Because of the highly complex and /or antagonistic nature of IFN response repertoire, the authors were not able to establish a protein/gene profile that predicts virus response and oncolytic activity. Although the results cannot be directly translated to clinical guidance , this is a promising approach for further refinement.

Few grammar/syntax comments are highlighted in the draft .

In future studies, group might choose to employ CRISR-Cas genetic screens for the unbiased detection of virus vulnerability genes.

Author's reply: We agree the sample population in the current study is very limited to achieve significance in terms of establishing a molecular profile predicting response to oncolytic viruses. Having the ultimate goal of fast monitoring the oncolytic potential of viruses capable of overcoming IFN-induced resistance of tumor cells, we have started with a pilot study addressing why patient-derived glioblastoma cultures can acquire increased resistance to oncolytic viruses in the presence of interferons and suggesting an approach to ranking glioblastoma cells by the acquired resistance. Our next step is to collect a more representative population of GBM samples, approaching clinical levels to continue these studies. We commented on it in the Discussions section (p.17 of the revised manuscript).

Round 2

Reviewer 1 Report

The authors provided additional information and addressed most of my concerns. 

However, as for my concern #3, the author's reply could not rule out the potential off-target effects. At least two individual siRNAs should be used to minimze the off-target effects and using siRNA resistant expression unit to rescue its knockdown effects is required to get its direct function in the filed.

Author Response

AUTHORS’ RESPONSE:

Dear Reviewer,

Thank you very much for your valuable comments. We agree that the usage of sufficient number of positive and negative controls is required to exclude all undesirable effects. We indicated limitations and requirements for further validation with other shRNAs and/or with the CRISPR-Cas9 approach in our future studies (Discussion and Conclusion sections, Line 678 and 707, respectively). When performing the shRNA knockdown experiments, we tested 2 to 3 different shRNA hairpins for each of target genes (shRNAs designed by Sigma-Aldrich and listed in their validated shRNA knockdown database were taken). The shRNAs showing the most efficient silencing of the target genes were selected for further usage.

Reviewer 1

Comments and Suggestions for Authors

The authors provided additional information and addressed most of my concerns.

However, as for my concern #3, the author's reply could not rule out the potential off-target effects. At least two individual siRNAs should be used to minimze the off-target effects and using siRNA resistant expression unit to rescue its knockdown effects is required to get its direct function in the filed.

Reviewer 2 Report

Many thanks for addressing our concerns. The manuscript, although still limited in its scientific impact,  incorporates a substantial amount of experimental work and can contribute to the greater understanding of the biology of oncolytic viruses.

Author Response

Reviewer 2

Comments and Suggestions for Authors

Many thanks for addressing our concerns. The manuscript, although still limited in its scientific impact, incorporates a substantial amount of experimental work and can contribute to the greater understanding of the biology of oncolytic viruses.

AUTHORS' RESPONSE:

Dear Reviewer,

Thank you very much for your positive response. We hope to increase the merit of our results for clinical applications by obtaining a more significant collection of glioblastoma samples.

Reviewer 3 Report

Lipatova and colleagues have made a great job improving the manuscript entitle "Multi-Omics analysis of glioblastoma cells' sensitivity to oncolytic viruses," however there are some concerns that have arisen after the changes and some other that were not properly answered.

The graphical abstract is not very serious and doesn’t represent the obtained results or the workflow of the paper. It is not clear what the authors what to show. It must be a summary in just one picture, and it doesn’t seem the case. The presence of a graphical abstract it is not mandatory in Cancers.

The GBM cell lines names have not been applied to all the cultures “However, IFNα has induced complete or partial protection of 5222 and 5522 cells, respectively. The 4114 cells remained virus-sensitive even upon treatment with IFNα, suggesting a severe deficiency of IFN response”

The idea of putting a letter to each volcano plot it is to refer specifically to them when the authors are explaining the results showed in those analysis. These could help to understand and follow the analysis of the results. No mention to Fig2 but a general one it shown in the text. Same for the supplementary figures.

Minor:

Some of the references appear in other type of style ( i.e [50], [33,49]).

Author Response

Dear Reviewer,

Thank you very much for your positive response. Below we describe revisions made to address the raised concerns.

REVIEWER 3

Comments and Suggestions for Authors

Lipatova and colleagues have made a great job improving the manuscript entitle "Multi-Omics analysis of glioblastoma cells' sensitivity to oncolytic viruses," however there are some concerns that have arisen after the changes and some other that were not properly answered.

The graphical abstract is not very serious and doesn’t represent the obtained results or the workflow of the paper. It is not clear what the authors what to show. It must be a summary in just one picture, and it doesn’t seem the case. The presence of a graphical abstract it is not mandatory in Cancers.

AUTHORS’ RESPONSE: We revised the graphical abstract to summarize the subjects, methods and results of the study. Doing that, we used a general template suggested by Elsevier (https://www.elsevier.com/authors/tools-and-resources/visual-abstract). If our current version does not properly fit your comment, we agree to delete TOC.

The GBM cell lines names have not been applied to all the cultures “However, IFNα has induced complete or partial protection of 5222 and 5522 cells, respectively. The 4114 cells remained virus-sensitive even upon treatment with IFNα, suggesting a severe deficiency of IFN response”

AUTHORS’ RESPONSE: Thank you, we accidentally missed that in the previous revisions. Done.

The idea of putting a letter to each volcano plot it is to refer specifically to them when the authors are explaining the results showed in those analysis. These could help to understand and follow the analysis of the results. No mention to Fig2 but a general one it shown in the text. Same for the supplementary figures.

AUTHORS’ RESPONSE: We revised referring to Figures with the emphasis on subfigures, where appropriate.

Minor:

Some of the references appear in other type of style ( i.e [50], [33,49]).

AUTHORS’ RESPONSE: Done.